# ON THE REFLECTION OF SENSITIVITY IN THE GENERALIZATION ERROR

## ABSTRACT

Even though recent works have brought some insight into the performance improvement of techniques used in state-of-the-art deep-learning models, more work is needed to understand the generalization properties of over-parameterized deep neural networks. We shed light on this matter by linking the loss function to the output's sensitivity to its input. We find a rather strong empirical relation between the output sensitivity and the variance in the bias-variance decomposition of the loss function, which hints on using sensitivity as a metric for comparing generalization performance of networks, without requiring labeled data. We find that sensitivity is decreased by applying popular methods which improve the generalization performance of the model, such as (1) using a deep network rather than a wide one, (2) adding convolutional layers to baseline classifiers instead of adding fully connected layers, (3) using batch normalization, dropout and max-pooling, and (4) applying parameter initialization techniques.

## 1 INTRODUCTION

In machine-learning tasks, the main challenge a network designer faces is to find a model that learns the training data and that is able to predict the output of unseen data with high accuracy. The first part is easily achievable in current over-parameterized deep neural networks. Yet the second part, referred to as generalization, demands careful expert hand-tuning (LeCun et al. (2015), Goodfellow et al. (2016)). Modern convolutional neural network (CNN) architectures that achieve state-of-the-art results in computer-vision tasks, such as ResNet (He et al., 2016) and VGG (Simonyan & Zisserman, 2014), attain high-generalization performance. Part of their success is due to recent advances in hardware and the availability of large amounts of data. However, among models applying these resources, some outperform others. Hence, the key questions about when and why some models generalize, still remain a mystery (Neyshabur et al., 2017).

In this paper, by investigating the link between sensitivity and generalization, we get one step closer to understanding the generalization properties of neural networks. Our findings suggest a relation between the sensitivity metric and the variance term in the bias-variance decomposition of the test loss (Geman et al. (1992), Tibshirani (1996), Neal et al. (2018)). This relation provides us with some intuition about the link, when the bias is small, between sensitivity and loss. We present strong empirical results on the link between the sensitivity and the test loss in well-established deep neural networks with ReLU (Nair & Hinton, 2010) non-linearity.

Building on this relation, we can use the sensitivity metric to examine which network is more prone to overfitting, and we rediscover the generalization properties of certain structures. We provide numerical evidence that a significant decrease in the network output's sensitivity to input perturbations can be a good predictor of the generalization capability of certain techniques used in state-of-the-art models, such as batch normalization (Ioffe & Szegedy, 2015) and He initialization (He et al., 2015). Sensitivity also supports the commonly accepted observation that depth is better than width in deep neural networks (Bengio & Delalleau (2011), Montufar et al. (2014), Telgarsky (2016), Mhaskar et al. (2017)). Furthermore, in certain settings, the sensitivity of untrained networks can be used as a proxy for the generalization error of trained networks.

## 2 RELATED WORK

Recently, Sokolić et al. (2017) suggested bounding the generalization error of deep neural networks with the spectral norm of the input-output Jacobian matrix, a measure of output sensitivity to its inputs. Empirical support for their conclusions is provided in Novak et al. (2018) through experiments on fully connected neural networks. Further research is needed to determine the conditions that are required to compare the generalization performance of models by using sensitivity metrics. For this purpose, we provide theoretical and empirical insights to the link between sensitivity and generalization by studying the relation between sensitivity and the variance term in the bias-variance decomposition of the loss function. To the best of our knowledge, this work is the first to find such a relation. In our work, we not only elaborate on the relation between sensitivity and loss for a wide range of networks (not limited to fully connected) but also empirically show that many state-of-the-art methods improve performance alongside reducing the sensitivity of the network.

To avoid overfitting in deep-learning architectures, regularization techniques are applied: for example, weight decay, early stopping, dropout (Srivastava et al., 2014), and batch normalization (BN) (Ioffe & Szegedy, 2015). Ioffe & Szegedy (2015) argues that the reason for the success of BN is that it addresses the internal-covariant-shift phenomenon. However, Santurkar et al. (2018) argues against this belief and explains that the success of BN is due to its ability to make the optimization landscape smoother. Here, we look at the success of dropout and BN from another perspective. These methods decrease the output sensitivity to random input perturbations in a same manner as decreasing the test loss, resulting in better generalization performance.

Designing neural network architectures is one of the main challenges in machine-learning tasks. One major line of work in this regard is comparing deep and shallow networks (Bengio & Delalleau (2011), Mhaskar et al. (2017), Wu et al. (2019), Ba & Caruana (2014), Montufar et al. (2014), and Simonyan & Zisserman (2014)). It is shown in Telgarsky (2016) that to approximate a deep network, a shallow network requires an exponentially larger number of units per layer. After finding a satisfactory architecture, the trainability of the network needs to be carefully assessed. In order to avoid exploding or vanishing gradients, Glorot & Bengio (2010) and He et al. (2015) introduce parameter initialization techniques that are widely used in current frameworks. By linking sensitivity and generalization, we present a new viewpoint on understanding the success of current state-of-the-art architectures and techniques.

Previous theoretical studies attempting to solve the mystery of generalization mostly include generalization error (GE) bounds that use complexity measures such as VC-dimension and Rademacher complexities (Xu & Mannor (2012), Kawaguchi et al. (2017), Arora et al. (2018), and Sokolić et al. (2017)). GE depends not only on the architecture of the network, but also on the structure of the data itself[1]. This is why the same network has a lower GE when trained with structured data than with random data (Zhang et al., 2016). Here, we do not study GE bounds; however, in order to find complexity measures to bound GE, we should take note of the link between sensitivity and generalization.

There has been research on sensitivity analysis in neural networks with sigmoid and tanh activation functions (Dimopoulos et al. (1995), Fu & Chen (1993), and Zeng & Yeung (2001)). Yang et al. (2013) introduce a sensitivity-based ensemble approach which selects individual networks with diverse sensitivity values from a pool of trained networks. Piche (1995) performs a sensitivity analysis in neural networks to determine the required precision of the weight updates in each iteration. In this work, we extend these results to networks with ReLU non-linearity with a different goal; to study the relation between sensitivity and generalization error in the state-of-the-art deep neural networks.

The metrics used for evaluating the network performance on training data cannot reflect generalization, because an over-parameterized model can achieve zero training loss for networks trained with randomly labeled data (Zhang et al., 2016). There have been recent attempts to predict the test loss for supervised-learning tasks (Novak et al. (2018), Jiang et al. (2018), and Wang et al. (2018)). Philipp & Carbonell (2018) introduces the non-linearity coefficient (NLC) as a gradient-based complexity measure of the neural network, which is empirically shown to be a predictor of the test error for fully connected neural networks. According to our results on both fully connected and convolutional neural networks, sensitivity is also a predictor of the test error, even before the networks are trained, which suggests sensitivity as a computationally inexpensive architecture-selection metric.

---

[1]In this work, we do not investigate the structure of the data set.

**Paper Outline.** We formally define loss and sensitivity metrics in Section 3.1 and Section 3.2, respectively. In Section 4, we state the main findings of the paper and present the numerical and analytical results supporting that. Later in Section 5, we propose a possible proxy for generalization properties of certain structures and certain methods. Finally in Section 6, we further discuss the observations followed up by a conclusion in Section 7. The empirical results presented in the paper are for an image-classification task on the CIFAR-10 dataset. In Appendix 8.6, we present the empirical results for MNIST and CIFAR-100 datasets. In Appendix 8.7, we present the empirical results for a regression task with the Boston housing dataset.

## 3 PRELIMINARIES

Consider a supervised-learning task, where the model predicts ground-truth output $y \in \mathcal{Y} := \mathbb{R}^K$ for an input $x \in \mathcal{X} := \mathbb{R}^D$. The predictor $F_\theta : \mathcal{X} \to \mathcal{Y}$ is a deep neural network parameterized by the parameter vector $\theta$ that is learned on the training dataset $\tau_t$ by using the stochastic learning algorithm $\mathcal{A}$. The training dataset $\tau_t$ and the testing dataset $\tau_v$ consist of i.i.d. samples drawn from the data distribution $p$. With some abuse of notation, we use $\sim$ when the samples are uniformly drawn both from a set of samples and from a probability distribution.

### 3.1 LOSS

Our main focus is a classification task where the loss function is the cross-entropy criterion. The average test loss can be defined as

$$L = \mathbb{E}_{\theta^*}[L_{\theta^*}] = \mathbb{E}_{\theta^*}\left[\mathbb{E}_{(x,y)\sim\tau_v}\left[-\sum_{k=1}^K y^k \log F_{\theta^*}^k(x)\right]\right], \tag{1}$$

where $\theta^*$ is the random[2] parameter vector found by $\mathcal{A}$ minimizing the train loss defined on $\tau_t$, $K$ is the number of classes and $F_{\theta^*}^k$ is the $k$-th entry of the vector $F_{\theta^*}$, which is the output of the softmax layer, i.e., $F_{\theta^*} = \text{softmax}(f_{\theta^*}(x))$, where $f_{\theta^*}(x)$ is the output of the last layer of the network. In classification tasks, the output space is $\mathcal{Y} := [0,1]^K$, and the output vector is the probability assigned to each class.

### 3.2 SENSITIVITY

Let us inject an external noise to the input of the network and compute the noise in the output. If the original input vector is $x \in \mathcal{X}$ and we add an i.i.d. normal noise vector $\varepsilon_x \sim \mathcal{N}(0, \sigma_{\varepsilon_x}^2 I)$ to the input, then the output noise due to the input noise $\varepsilon_x \in \mathcal{X}$ is $\varepsilon_y = f_\theta(x + \varepsilon_x) - f_\theta(x) \in \mathcal{Y}$. We use the variance of the output noise[3], averaged over its $K$ entries, as a measure of sensitivity: $S_\theta = \text{Var}(\overline{\varepsilon_y})$. The average sensitivity is

$$S = \mathbb{E}_\theta[S_\theta] = \mathbb{E}_\theta\left[\text{Var}(\overline{\varepsilon_y})\right] = \mathbb{E}_\theta\left[\text{Var}_{x,\varepsilon_x}\left[\frac{1}{K}\sum_{k=1}^K \varepsilon_y^k\right]\right], \tag{2}$$

where $\varepsilon_y^k$ is the $k$-th entry of the vector $\varepsilon_y$. In the following sections, we use $S_{\text{before}} = \mathbb{E}_\theta[S_\theta]$ and $S_{\text{after}} = \mathbb{E}_{\theta^*}[S_{\theta^*}]$ when the expectation is over the network parameters before and after training, respectively. We consider unspecific sensitivity (the average over the entries of the output noise), which requires unlabeled data samples, as opposed to specific sensitivity (the sensitivity of the output of the desired class) (Tartaglione et al., 2018). By expanding Equation (2) up to the first order, the sensitivity $S$ is approximated by the product of the variance of the input noise vector $\sigma_{\varepsilon_x}^2$ and the square of the Frobenius norm of the Jacobian of the output of the network (Novak et al., 2018). In experiments, we prefer $S$ to the Jacobian, because in order to compute $S$ it is enough to look at the network as a black box that given an input, generates an output, without requiring further knowledge of the model.

---

[2]The randomness is introduced by the stochastic optimization algorithm $\mathcal{A}$ and the randomized parameter initialization technique.

[3]Note that we consider the output of the neural network and not the output of the softmax layer. The softmax layer can be interpreted as a normalizing layer pushing the values to be between 0 and 1.

## 4    SENSITIVITY VS. LOSS

### 4.1    MAIN OBSERVATION

After training, an ideal predictor is expected to have a good generalization ability. An ideal predictor should also be robust; given similar inputs, the outputs should be close to each other. Assuming that the unseen data is drawn from the same distribution as the training data, there should be a link between the two concepts of robustness and generalization. Robustness here is the average-case robustness, not the worst-case robustness (adversarial robustness). We measure the robustness by computing $S$ (Equation (2)), and considering near-zero train loss, we refer to the test loss $L$ (Equation (1)) as the generalization error. According to our observations on a wide set of experiments, we find a rather strong relation between the sensitivity $S$ and the generalization error $L$. State-of-the-art networks decrease the generalization error alongside with the sensitivity of the output of the network with respect to the input (Figure 1).

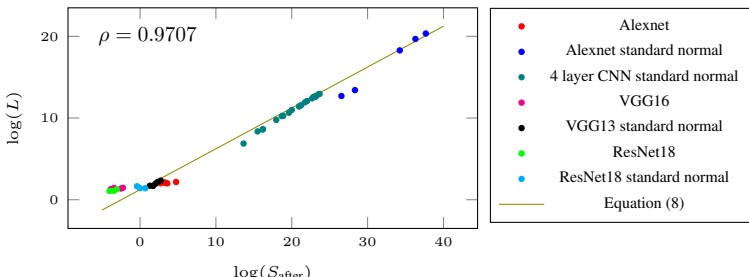

Figure 1: Sensitivity $S_{\text{after}}$ versus test loss $L$ for popular CNN architectures. The parameter initialization is Xavier (Glorot & Bengio, 2010) with uniform distribution unless stated as standard normal distribution. The networks are trained and fully converged on a subset of the CIFAR10 training dataset and are evaluated on the entire CIFAR10 testing dataset. Each point indicates a network with different numbers of channels and hidden units, and is averaged over multiple runs. For more details on configurations refer to Appendix 8.1. The Pearson correlation coefficient $\rho$ between data points is indicated in the figure.

### 4.2    NUMERICAL EXPERIMENTS

Many factors influence the generalization performance of deep-learning models, among which network topology, initialization technique, and regularization method. We study the influence of these three factors on the sensitivity $S$ and keep all the other factors, including the learning algorithm, the same. We refer to fully connected neural networks as FC and convolutional neural networks as CNN. First, we examine the relation between $S_{\text{after}}$ and $L$ for different architectures, different initialization techniques, and different regularizations. We observe a strong link between $S_{\text{after}}$ and $L$; a lower sensitivity has a lower test loss (Figures 2 and 3). This strong link suggests that, by comparing the sensitivity $S_{\text{after}}$, we can simply compare generalization performances of multiple neural networks that are fully trained (with various hyper-parameters, types of layer and regularization methods). Using sensitivity as a proxy for test loss is particularly advantageous in settings where labeled training data is limited; assessing generalization performance can be then done without having to sacrifice training data for the validation set. Next, when initialization is fixed to the standard normal distribution and no explicit regularization technique is applied, we investigate the link between $S_{\text{before}}$ and $L$ (Figure 4 in Section 6.1). We see again a clear relation between $S_{\text{before}}$ and $L$; $S_{\text{before}}$ can potentially be used as an architecture-selection metric before training the models.

### 4.3    BIAS-VARIANCE DECOMPOSITION

In this section, a crude approximate relation between sensitivity and generalization error is established through the link between sensitivity and the variance term in the bias-variance decomposition of the mean square error. First, we find the link between the cross-entropy loss and the mean square error. Then, we present the relation between sensitivity and the variance term in the bias-variance decomposition of the mean square error. Finally, the link between sensitivity $S$ and generalization error $L$ is established.

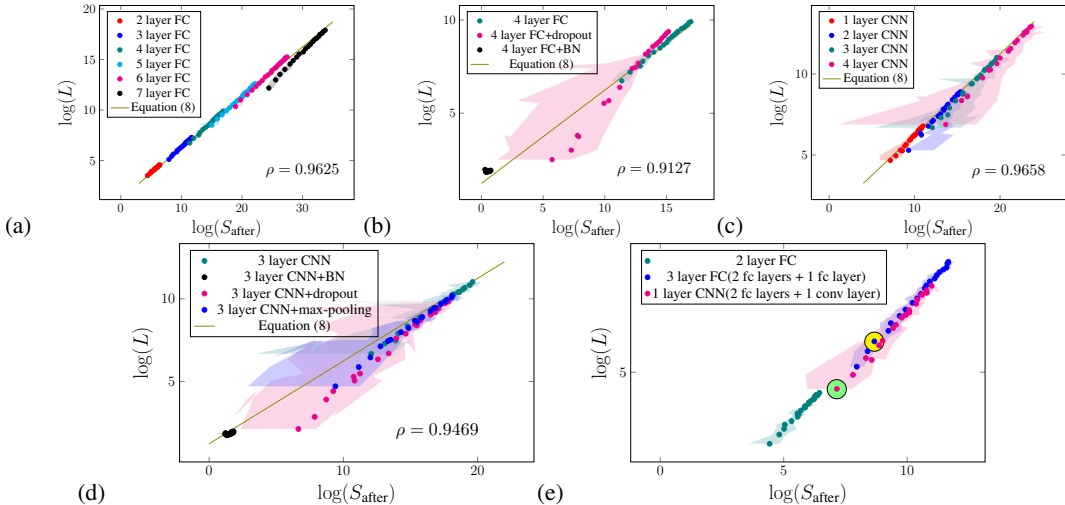

Figure 2: Test loss versus sensitivity for networks trained on a subset of the CIFAR-10 training dataset where the network parameters are initially drawn from a standard normal distribution. Each point indicates a network with different numbers of channels and hidden units and is the average over multiple runs. The interval indicates the minimum and maximum values of sensitivity over multiple runs. **(a)** Fully connected neural networks. **(b)** 4-layer FC trained with or without regularization. **(c)** Convolutional neural networks. **(d)** 3-layer CNN trained with or without regularization. **(e)** Comparison between adding a convolutional layer and adding a fully connected layer to a baseline classifier that is a fully connected neural network with one hidden layer.

When the predictor $F_{\theta*}(x)$ assigns the probability $F^c_{\theta*}(x)$ to the correct class $c$ and $1 - F^c_{\theta*}(x)$ to another class (see Appendix 8.3 for details), the cross-entropy loss $L$ can be approximated as

$$L \approx \mathbb{E}_{(x,y,\theta*)} \left[ \frac{1}{\sqrt{2}} \sqrt{\sum_{k=1}^{K} \left( F^k_{\theta*}(x) - y^k \right)^2} \right]. \qquad (3)$$

We roughly approximate the right-hand side in Equation (3) with $\sqrt{L_{\text{MSE}}/2}$, where $L_{\text{MSE}}$ is the mean square error (MSE) criteria and is defined as

$$L_{\text{MSE}} = \mathbb{E}_{\theta*}[L_{\theta*\text{MSE}}] = \mathbb{E}_{\theta*} \left[ \mathbb{E}_{(x,y)\sim\tau_v} \left[ \|F_{\theta*}(x) - y\|^2 \right] \right]. \qquad (4)$$

Consider the classic notion of bias-variance decomposition for the MSE loss (Geman et al. (1992)), where the generalization error is the sum of three terms: bias, variance and noise, i.e., $L_{\text{MSE}} = \varepsilon_{\text{bias}} + \varepsilon_{\text{variance}} + \varepsilon_{\text{noise}}$. In this work, we consider the labels to be noiseless and neglect the third term $\varepsilon_{\text{noise}}$. The bias term is formally defined as

$$\varepsilon_{\text{bias}} = \mathbb{E}_{x,y} \left[ \|\mathbb{E}_{\theta*}[F_{\theta*}(x)] - y\|^2 \right],$$

and the variance term is formulated as

$$\varepsilon_{\text{variance}} = \sum_{k=1}^{K} \mathbb{E}_x \left[ \text{Var}_{\theta*}(F^k_{\theta*}(x)) \right]. \qquad (5)$$

Let us now draw an again crude approximate relation between $\varepsilon_{\text{variance}}$ and $S$ under strong assumptions on the probability distributions of $\theta$, $x$, and $\varepsilon_x$ (refer to Appendix 8.2 for more details). Given a feed-forward neural network with $M$ hidden layers and $H_l, 1 \leq l \leq M$, units per layer, where the non-linear activation function is positive homogeneous[4] with parameters $\alpha$ and $\beta$ (Equation (10) in Appendix 8.2), we have

$$\varepsilon_{\text{variance}} \approx \left( \frac{K-1}{K} \right) \left( S \cdot \frac{\sigma^2_x}{\sigma^2_{\varepsilon_x}} + \chi \right), \qquad (6)$$

[4]ReLU is a positive homogeneous function with $\alpha = 1$ and $\beta = 0$.

where

$$\chi = \frac{1}{K} \sum_{l=1}^{M} \sigma_{b_l}^2 \prod_{i=l+1}^{M} \left( \frac{\alpha^2 + \beta^2}{2} \right) \sigma_{w_i}^2 H_i, \tag{7}$$

and $K$ is the number of units in the output of the softmax layer and $\sigma_{w_l}^2$, $\sigma_{b_l}^2$, $\sigma_x^2$, and $\sigma_{\varepsilon_x}^2$ are the second moment of weights and biases of layer $l$, input $x$ and input noise $\varepsilon_x$, respectively. Equation (7) can be extended to convolutional neural networks by replacing $H_i$ with $fan_{in}$ of weights of layer $i$.

Given an infinite amount of training data, the bias represents the best performance of our model which can be approximated by the training loss (Mehta et al. (2019), Ng, Fortmann-Roe (2012)). In our experiments, the training loss is close to zero, hence if we neglect the bias term $\varepsilon_{\text{bias}}$ in the test loss we have,

$$L \approx \sqrt{\frac{1}{2} \left( \frac{K-1}{K} \right) \left( S \cdot \frac{\sigma_x^2}{\sigma_{\varepsilon_x}^2} + \chi \right)}, \tag{8}$$

where $\chi$ is given by Equation (7). In experiments, we observe that $\sigma_{b_l}^2$ is usually very small or zero (for instance in ResNets $b_l = 0$), making $\chi \approx 0$.

According to Equation (6) and the relation between $L_{\text{MSE}}$ and $L$, to compare networks with the small value of $\varepsilon_{\text{bias}}$ (which is usually the case in deep neural networks where the bias is approximated with the near-zero training loss), the test loss can be approximated by the sensitivity following Equation (8). Despite the strong assumptions and crude approximations to get Equation (8), the numerical experiments show a rather surprisingly good match with Equation (8) (Figures 1, 2 and 3), when $\chi$ is neglected in Equation (8). It is interesting to note that the right-hand side of Equation (8) is computed without requiring labeled data points.

If $\varepsilon_{\text{bias}}$ can no longer be approximated by the training loss, which may in part explain the poorer match in lower values of Figure 1, we need more training data to make this approximation valid. In our experiments, we train the networks with only a subset of the training dataset, we also show in Appendix 8.8 that by training with more data samples, numerical results become closer to Equation (8).

## 5    SENSITIVITY AS A PROXY FOR GENERALIZATION

In this section, we argue that methods improving the generalization performance of neural networks remarkably reduce the sensitivity $S$. In particular, we revisit some common notions regarding generalization in deep neural networks: adding convolutional layers instead of fully connected layers, using deeper networks instead of wider ones, adding dropout, batch normalization (BN) and max-pooling layers, and initializing the network parameters by techniques such as Xavier and He.

### 5.1    COMPARING DIFFERENT ARCHITECTURES

**Convolutional vs Fully Connected Layers.** The relation between the sensitivity $S$ and the generalization error $L$ supports the common view that CNNs outperform FCs in image-classification tasks. We empirically observe that, given a CNN and an FC with the same number of parameters, the CNN has lower sensitivity and test loss than the FC. Moreover, some CNNs with more parameters than FCs have both lower sensitivity and lower test loss, even though they are more over-parameterized. Let us start from a baseline classifier with one hidden layer (2 layers in total), and we compare the effect of adding another fully connected layer versus adding a convolutional layer in Figure 2(e). We vary the number of parameters of 1-layer CNNs (which consist of 2 fc layers and 1 conv layer) from 450k to 10M by increasing the number of channels and hidden units, whereas the number of parameters for 3-layer FCs varies from 320k to 1.7M. Despite the large number of parameters of CNNs, they suffer from less overfitting and have a lower sensitivity $S$ than FCs. Next, let us compare a FC to a CNN with the same number of parameters in Figure 2(e): a 3-layer FC with 140 units in each layer (yellow mark) and a 1-layer CNN with 5 channels and 100 units (green mark), both have 450k parameters. The CNN has remarkably lower sensitivity and test loss than the FC, which indicates better performance compared to the FC with the same number of parameters.

**Depth vs Width.** Consider a feedforward FC with ReLU activation function where all the network parameters follow the standard normal distribution and are independent from each other and from the

input. If we have $M$ layers with $H$ units in each hidden layer, $K$ units in the output layer and the input layer with $D$ units, then (see Appendix 8.4 for details)

$$S = \frac{D}{K} \left( \frac{H}{2} \right)^M \sigma_{\varepsilon_x}^2.$$
(9)

According to Equation (9), considering two neural networks with the same value for $H^M$, one being deep and narrow (higher $M$ and lower $H$), and the other being shallow and wide (lower $M$ and higher $H$), subject to the same noise level $\sigma_{\varepsilon_x}^2$, the deeper network has lower sensitivity $S$. Assuming both networks have near-zero training losses, then, depth is better than width regarding generalization in fully connected neural networks. The empirical results in Figure 2(a) support Equation (9). For instance, a 4-layer FC with 500 units per layer (the last point among 4-layer FCs), has the same value for $(\frac{H}{2})^M$ as a 5-layer FC with 165 units per layer (the 4th point among 5-layer FCs). In Figure 2(a), these two networks have the same value for both *sensitivity* and *loss*, and all narrower 5-layer networks (with 100, 120, and 140 units) have better performance than the wide 4-layer network (with 500 units). Note that Equation (9) only holds for networks with parameters drawn from the standard normal distribution and does not hold when the parameter initialization is Xavier or He.

## 5.2 REGULARIZATION AND INITIALIZATION METHODS

Figures 2(b) and 2(d) show the sensitivity $S_{\text{after}}$ versus the test loss $L$, for different regularization methods. In particular, we study the effect of dropout and BN on the sensitivity in the FCs; and we apply dropout, BN and max-pooling for the CNNs. The results are consistent with the relation between sensitivity $S$ and loss $L$. For all mentioned regularization techniques, we observe a shift of the points towards the bottom left; this shift suggests that these techniques improve generalization by decreasing the network sensitivity to input perturbations. This is especially noticeable in the BN case, where both the sensitivity and test loss decrease dramatically. This suggests that batch normalization improves performance by making the network less sensitive to input perturbations.

Another interesting observation is the effect of various parameter initialization techniques on the sensitivity and loss values, after the networks are trained (Figure 3). We consider four initialization techniques for network parameters in our experiments: (i) Standard Normal distribution (SN), (ii) Xavier (Glorot & Bengio, 2010) initialization method with uniform distribution (XU), (iii) He (He et al., 2015) initialization method with uniform distribution (HU), and (iv) He initialization method with normal distribution (HN). As shown in Figure 3, the relation between sensitivity $S_{\text{after}}$ and test loss $L$ provides us a new view-point of the success of the state-of-the-art initialization techniques; the He initialization method with normal distribution has the best generalization performance, alongside the lowest sensitivity value.

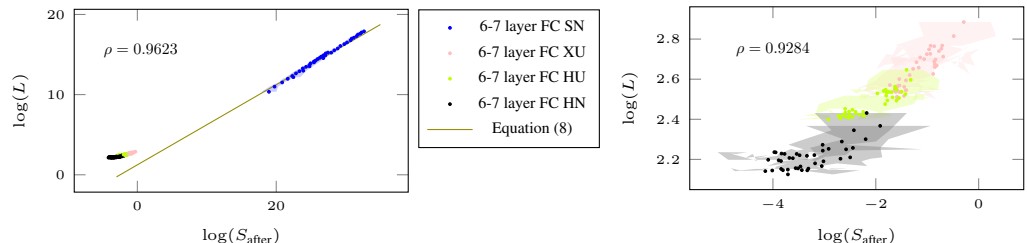

Figure 3: Test loss versus sensitivity for networks trained on a subset samples of the CIFAR-10 dataset where networks are initialized with different methods. On the right, we have a zoom in plot of the left figure.

## 6 DISCUSSION

### 6.1 SENSITIVITY OF UNTRAINED NETWORKS AS A PROXY FOR THE TEST LOSS

A similar trend, as in Section 4, is observed for neural networks that are not yet trained. In Figure 4, the sensitivity $S_{\text{before}}$ is measured before the networks are trained, and the test loss $L$ is measured after

the networks are trained on a subset of the training dataset. The parameters in the fully connected and convolutional networks are initialized by sampling from the standard normal distribution, and no explicit regularization (dropout, BN, and max-pooling) is used in the training process. These two conditions are necessary, because regularization techniques only affect the training process, hence $S_{\text{before}}$ is the same for networks with or without regularization layers, and the He and Xavier initialization techniques force the sensitivity to be fixed regardless of the number of units in hidden layers. Hence, under these two conditions, the generalization performance of untrained networks with different architectures can be compared. The strong link between the sensitivity of untrained networks $S_{\text{before}}$ and the test loss $L$ suggests that generalization of neural networks can be compared before the networks are even trained, making sensitivity a computationally inexpensive architecture-selection method.

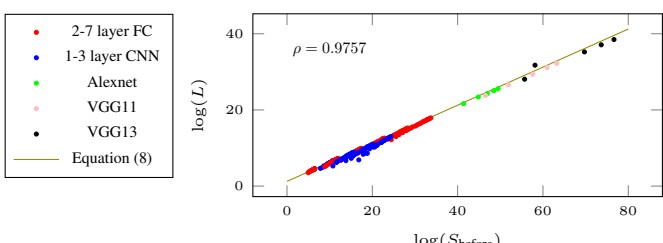

Figure 4: Test loss of trained models versus sensitivity of untrained models for networks whose parameters are initially drawn from the standard normal distribution. Note that the regularization techniques BN, dropout and max-pooling are removed from Alexnet, VGG11, and VGG13 configurations.

## 6.2 FINAL REMARKS

It is important to note that sensitivity is not the only factor affecting generalization. The sensitivity metric does not address the challenges brought by Zhang et al. (2016); the sensitivity of a random-labeled input is the same as the sensitivity of a true-labeled input, with very different loss values. The pixel-wise linear input perturbations, considered in our experiments, might not be realistic; ideally, we would like to perturb the input in the latent space of the generative model of the input image. Also, the relation between $S$ and $L$, requires the non-linearity to be positive homogeneous, hence the generalization properties of networks with sigmoid and tanh activation functions remain unexplained. It is not possible to find a single metric to explain the generalization properties across a space as large as deep neural networks and across the non-convex landscapes of deep-learning problems. But the sensitivity metric presented in this work is strongly related to generalization, and even though Equation (8) was derived under very strong assumptions and crude approximations, it is surprisingly quite close to the empirical results in a wide set of settings.

The sensitivity metric can be affected by re-scaling. If we re-scale all the indices of the very last layer of the neural network, the accuracy stays the same as the network predicts the same class, however, both the test loss and sensitivity are multiplied by the square of the scaling factor. Therefore, the relation highlighted in this paper between sensitivity and generalization error cannot always be extended to a relation between sensitivity and accuracy. Also note that, if we re-scale the input $x$, define a new input $x_{\text{new}} = \alpha x$, and de-scale the output $f_{\text{new}}(\cdot) = f(\cdot)/\alpha$, then due to homogeneity of $f$, we have $L_{\text{new}} = L$, however, $S_{\text{new}} = S/\alpha^2$. Hence sensitivity cannot replace test loss and to compare the generalization error of models by using their sensitivity, the comparison needs to be made fairly; the input of both networks should be drawn from the same distribution.

## 7 CONCLUSION

We find that the sensitivity metric is a strong indicator of overfitting. Given multiple networks, with near-zero training loss, to choose from with various hyper-parameters, the best architecture appears to be the one with the lowest sensitivity value. Sensitivity can potentially be used as an early stopping criterion and as an architecture-selection method; given multiple networks to choose from by only computing the sensitivity to the given input, the best model is the one with the lowest sensitivity. One of the advantages of the sensitivity metric is that it can provide a loose prediction of the test loss without the use of any labeled data. This is especially important in applications where generating labels requires expensive experiments.

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

## 8 APPENDIX

### 8.1 EXPERIMENTAL DETAILS

The CIFAR-10 dataset[5] is used for the experiments presented in the paper. The fully connected neural networks have the same number of units in the hidden layers, varying from 100 to 500 with a step size of 20. For the convolutional neural networks the number of channels in convolutional layers vary from 5 to 25 with a step size of 5 (note that each time a channel is added in the convolutional layers, an extra 20 units is added to the fully connected layers of the CNN accordingly). As it is computationally expensive to reach zero training loss for the entire dataset, we choose a subset of the training set containing 1000 samples of the CIFAR-10 dataset. Zero training loss is necessary for a fair comparison between different networks since we would like to have the same value for $\varepsilon_{\text{bias}}$ and $\varepsilon_{\text{noise}}$ among them. For the optimization algorithm, we choose the Adam optimizer with its default parameters and we initialize the weights and biases with random values drawn from a distribution according to the initialization technique stated in each figure. The nonlinearity is set to be the ReLU function throughout the experiments. We stop the training when the training loss reaches below the threshold $10^{-5}$ for 10 times. In case this condition is not met, we stop the training after 2000 epochs (each epoch is iterations over the mini-batches of the entire training set). The noise added to the input image is a random tensor with the same size as the input and is drawn from the Gaussian distribution with zero mean and 0.1 standard deviation. The output noise is first averaged over all its $K$ entries (for CIFAR-10 the number of classes is $K = 10$), then we take its variance over inputs of the testing dataset and the input noise, finally it is averaged over multiple runs.

Using the notations:

- Conv(number of filters, kernel size, stride, padding)
- Maxpool(kernel size)
- Linear(number of units)
- Dropout(dropout rate)

for layers of a convolutional neural network where Conv and Linear layers also include the ReLU non-linearity except the very last linear layer. The configurations that are used are:

- The Alexnet (Krizhevsky et al., 2012): Conv(h, 3, 2, 1) - Maxpool(2) - Conv(3*h, 3, 1, 1) - Maxpool(2) - Conv( 6*h, 3, 1, 1) - Conv(4*h, 3, 1, 1) - Conv(4*h, 3, 1, 1) - Maxpool(2) - Dense layer - Dropout(0.5) - Linear(4096) - Dropout(0.5) - Linear(4096) - Linear(K)
  where h $\in [16, 32, 48, 64, 80]$
- VGG13 (Simonyan & Zisserman, 2014) : 2 x Conv(64*s, 3, 1, 1) - Maxpool(2) - 2 x Conv(128*s, 3, 1, 1) - Maxpool(2) - 2 x Conv(256*s, 3, 1, 1) - Maxpool(2) - 2 x Conv(512*s, 3, 1, 1) - Maxpool(2) - 2 x Conv(512*s, 3, 1, 1) - Maxpool(2) - Avgpool(2) - Dense layer - Linear(K)
  where s $\in [0.25, 0.5, 1, 1.5, 2]$ and all Conv layers have batch normalization
- Each block of a ResNet (He et al., 2016) configuration: 2 x Conv(h, 3, 1, 1) + Conv(h, 1, 1, 1) which Conv layers include BN and ReLU and the result of the summation goes into a ReLU layer and h is the number of channels.

VGG16 is the same as VGG13 with the difference that it has three layers in the last three blocks. VGG11 configuration is the same as tVGG13 except that in the first and second block it has one convolutional layer instead of 2. VGG19 is the same as VGG13 except that there is 4 conv layers instead of 2 in the last three blocks. ResNet18 has 2 blocks with h=64*s, 2 blocks with h=128*s, 2 blocks with h=256*s, and 2 blocks with h=512*s where s $\in [0.25, 0.5, 1, 1.5, 2]$. ResNet34 has 3 blocks with h=64*s, 4 blocks with h=128*s, 6 blocks with h=256*s, and 3 blocks with h=512*s where s $\in [0.25, 0.5, 1, 1.5, 2]$.

### 8.2 COMPUTATION OF EQUATION (6)

Computations of this section do not depend on the stage of the training, hence $\theta$ denotes the parameter vector at any stage of training. Let us recall the sensitivity metric (Equation (2)) definition

$$S = \mathbb{E}_{\theta}\left[\text{Var}_{x,\varepsilon_x}\left[\overline{\varepsilon_y}\right]\right],$$

where $\overline{\varepsilon_y} = 1/K \sum_{k=1}^{K} \varepsilon_y^k$ and $\varepsilon_y^k$ is the $k$-th entry of output noise vector $\varepsilon_y$ and is

$$\varepsilon_y^k = f_\theta^k(x + \varepsilon_x) - f_\theta^k(x) \cong \varepsilon_x \cdot \nabla_x^\top f_\theta^k(x),$$

where we apply a first order Taylor expansion of the output. For a one hidden layer neural network with $D$ input units, $H$ hidden units, and $K$ output units, we have $\theta = \{w_1 \in \mathbb{R}^{D \times H}, w_2 \in \mathbb{R}^{H \times K}, b_1 \in \mathbb{R}^H, b_2 \in \mathbb{R}^K\}$

---

[5]https://www.cs.toronto.edu/~kriz/cifar.html

where $w_l$ and $b_l$ are the weights and biases of layer $l$ ($l = 1$ is the hidden layer and $l = 2$ is the output layer) which are independently drawn from a zero-mean normal distribution: $w_1 \sim \mathcal{N}(0, \sigma_{w_1}^2 I)$, $w_2 \sim \mathcal{N}(0, \sigma_{w_2}^2 I)$, $b_1 \sim \mathcal{N}(0, \sigma_{b_1}^2 I)$, and $b_2 \sim \mathcal{N}(0, \sigma_{b_2}^2 I)$ (which has been studied in Bellido & Fiesler (1993)). We have

$$f_\theta^k(x) = \sum_{h=1}^{H} w_2^{hk} a(p^h) + b_2^k,$$

where $w_l^{jk}$ is the weight connecting unit $j$ in layer $l$ to unit $k$ in layer $l + 1$, $b_l^h$ is the bias term added to unit $h$ in layer $l + 1$, $p^h$ is the output of the linear transformation in the hidden unit $h$, i.e.,

$$p^h = \sum_{d=1}^{D} w_1^{dh} x^d + b_1^h,$$

and the non-linear activation function $a(\cdot)$ is a positive homogeneous function of degree 1; i.e.,

$$a(x) = \begin{cases} \alpha x & x > 0, \\ \beta x & \text{otherwise}, \end{cases} \tag{10}$$

where $\alpha$ and $\beta$ are non-negative hyper-parameters. ReLU follows Equation (10) with $\alpha = 1$ and $\beta = 0$. By applying the chain rule we obtain

$$\varepsilon_y^k = \sum_{d=1}^{D} \varepsilon_x^d \frac{\partial f_\theta^k(x)}{\partial x^d}$$

$$= \sum_{d=1}^{D} \varepsilon_x^d \sum_{h=1}^{H} w_2^{hk} w_1^{dh} \frac{\partial a(p^h)}{\partial p^h}.$$

Therefore, we have

$$\overline{\varepsilon_y} = \frac{1}{K} \sum_{k=1}^{K} \sum_{h=1}^{H} \sum_{d=1}^{D} \varepsilon_x^d w_2^{hk} w_1^{dh} \frac{\partial a(p^h)}{\partial p^h}.$$

The network parameters are assumed to be independent, and it is assumed that $x \perp\!\!\!\perp \theta$, and $\varepsilon_x \perp\!\!\!\perp \{\theta, x\}$. Moreove, the entries of the input vector $x$ are independent from each other with the same second moment, i.e., $\sigma_x^2 = \mathbb{E}[(x^d)^2]$ for $1 \le d \le D$. Consider the input noise $\varepsilon_x$ to be a vector of zero mean random variables, hence $S = \mathbb{E}_{\theta, x, \varepsilon_x}[(\overline{\varepsilon_y})^2]$. Then sensitivity becomes

$$S = \frac{1}{K^2} \sum_{k=1}^{K} \sum_{h=1}^{H} \sum_{d=1}^{D} \mathbb{E}_{\varepsilon_x}[(\varepsilon_x^d)^2] \mathbb{E}_{\theta, x} \left[ (w_2^{hk})^2 (w_1^{dh})^2 \left( \frac{\partial a(p^h)}{\partial p^h} \right)^2 \right]$$

$$= \frac{1}{K^2} \sum_{k=1}^{K} \sum_{h=1}^{H} \sum_{d=1}^{D} \sigma_{\varepsilon_x}^2 \sigma_{w_2}^2 \sigma_{w_1}^2 \frac{\alpha^2 + \beta^2}{2}, \tag{11}$$

where the second equation is followed by computing the expectation for zero-mean normal parameters. Let

$$var = \mathbb{E}_x \left[ \text{Var}_\theta[out] \right], \tag{12}$$

where $out = 1/K \sum_{k=1}^{K} f_\theta^k(x)$. Because of the homogeneity of the non-linearity $a(\cdot)$, we have $a(p^h) = p^h \cdot \frac{\partial a(p^h)}{\partial p^h}$. Hence

$$out = \frac{1}{K} \sum_{k=1}^{K} \left[ \sum_{h=1}^{H} w_2^{hk} \left( \sum_{d=1}^{D} w_1^{dh} x^d + b_1^h \right) \frac{\partial a(p^h)}{\partial p^h} + b_2^k \right].$$

Because the parameters are zero-mean, $var = \mathbb{E}_{\theta, x}[out^2]$ and we have

$$var = \frac{1}{K^2} \sum_{k=1}^{K} \sum_{h=1}^{H} \sum_{d=1}^{D} \mathbb{E} \left[ (x^d)^2 (w_2^{hk})^2 (w_1^{dh})^2 \left( \frac{\partial a(p^h)}{\partial p^h} \right)^2 \right]$$

$$+ \frac{1}{K^2} \sum_{k=1}^{K} \sum_{h=1}^{H} \mathbb{E} \left[ (w_2^{hk})^2 \right] \mathbb{E} \left[ (b_1^h)^2 \left( \frac{\partial a(p^h)}{\partial p^h} \right)^2 \right] + \frac{1}{K^2} \sum_{k=1}^{K} \mathbb{E} \left[ (b_2^k)^2 \right]$$

$$= \frac{1}{K^2} \sum_{k=1}^{K} \sum_{h=1}^{H} \sum_{d=1}^{D} \sigma_x^2 \sigma_{w_2}^2 \sigma_{w_1}^2 \frac{\alpha^2 + \beta^2}{2}$$

$$+ \frac{1}{K^2} \sum_{k=1}^{K} \sum_{h=1}^{H} \sigma_{w_2}^2 \sigma_{b_1}^2 \frac{\alpha^2 + \beta^2}{2} + \frac{1}{K^2} \sum_{k=1}^{K} \sigma_{b_2}^2,$$

which is followed by taking the expectations over the parameters with zero-mean normal distributions. Therefore, we obtain

$$var = S \cdot \frac{\sigma_x^2}{\sigma_{\varepsilon_x}^2} + \frac{H}{K}\sigma_{w_2}^2\sigma_{b_1}^2\frac{\alpha^2 + \beta^2}{2} + \frac{\sigma_{b_2}^2}{K},$$

where $\sigma^2$ is the notation used for the second moment of a random variable. Following the same computations for a neural network with $M$ hidden layers, we have

$$var = S \cdot \frac{\sigma_x^2}{\sigma_{\varepsilon_x}^2} + \frac{1}{K}\sum_{l=1}^{M}\sigma_{b_l}^2\prod_{i=l+1}^{M}\frac{\alpha^2 + \beta^2}{2}\sigma_{w_i}^2 H_i, \tag{13}$$

where $K$ is the number of units of the output layer $M + 1$. We refer to second term in the right-hand side of Equation (13) as $\chi$. Its value is a very rough approximation given the numerous assumptions made above, but in practice it can often be neglected because $\sigma_{b_l}^2$ is very small or zero (the ResNet configurations do not have biases) in most of our experiments.

The first order Taylor expansion for an arbitrary function at the average of the input is $g(x) \approx g(\mathbb{E}[x]) + g'(\mathbb{E}[x])(x - \mathbb{E}[x])$. Taking the variance of this equation we have

$$\text{Var}(g(x)) \approx \left(g'(E[x])\right)^2 \text{Var}(x).$$

Here the function $g(\cdot)$ is the softmax function

$$F_\theta^k(x) = \frac{\exp(f_\theta^k(x))}{\sum_{i=1}^K \exp(f_\theta^i(x))}.$$

The input of the softmax function is a $K$-dimensional vector, so the first order Taylor expansion includes the vector-matrix multiplication of the covariance and the input vector. We assume the outputs of the last layer are independent from each other, so the covariance matrix is considered a diagonal matrix. Because the parameters are considered to be zero-mean, the input of the softmax has zero mean, $\mathbb{E}[f_\theta^k(x)] = 0$ for $1 \leq k \leq K$, then

$$\text{Var}(F_\theta^k(x)) \approx \sum_{i=1}^K \left(\frac{\partial F_\theta^i(x)}{\partial f_\theta^i(x)}\right)^2 \text{Var}(f_\theta^i(x))$$

$$\approx \left(\frac{1}{K}\cdot\left(1 - \frac{1}{K}\right)\right)^2 \text{Var}(f_\theta^k(x)) + \left(\frac{1}{K^2}\right)^2 \sum_{\substack{i=1\\i\neq k}}^K \text{Var}(f_\theta^i(x)),$$

since softmax$(0) = 1/K$. Therefore,

$$\varepsilon_{\text{variance}} = \sum_{k=1}^K \mathbb{E}_x\left[\text{Var}(F_\theta^k(x))\right] \approx \left(\frac{(K-1)^2}{K^4} + \frac{K-1}{K^4}\right)\cdot K^2 var = \left(\frac{K-1}{K}\right)\left(S\cdot\frac{\sigma_x^2}{\sigma_{\varepsilon_x}^2} + \chi\right),$$

which completes the computations.

## 8.3 THE RELATION BETWEEN THE CROSS ENTROPY LOSS AND THE MEAN SQUARE ERROR

We rewrite the cross-entropy loss (Equation (1)) as

$$L = \mathbb{E}_{\theta^*}[L_{\theta^*}] = \mathbb{E}_{x,c,\theta^*}\left[-\log(F_{\theta^*}^c)\right],$$

where $1 \leq c \leq K$ is the index of the true class for the input $x$, i.e., $y^c = 1$ and $y^k = 0$ for $k \neq c$. For simplicity we use the notation $F_{\theta^*}^c$ instead of $F_{\theta^*}^c(x)$ in this section. For the MSE loss we have

$$L_{\text{MSE}} = \mathbb{E}_{x,y,\theta^*}\left[\sum_{k=1}^K \left(F_{\theta^*}^k - y^k\right)^2\right].$$

Because $\sum_{k=1}^K F_{\theta^*}^k = F_{\theta^*}^c + \sum_{\substack{j=1\\j\neq c}}^K F_{\theta^*}^j = 1$ the summation inside the above expectation can be rewritten as

$$\sum_{k=1}^K \left(F_{\theta^*}^k - y^k\right)^2 = (1 - F_{\theta^*}^c)^2 + \sum_{\substack{j=1\\j\neq c}}^K \left(F_{\theta^*}^j\right)^2$$

$$= (1 - F_{\theta^*}^c)^2 + (1 - F_{\theta^*}^c)^2 - \sum_{\substack{i=1\\i\neq c}}^K\sum_{\substack{j=1\\j\neq i,c}}^K F_{\theta^*}^i F_{\theta^*}^j$$

$$= 2(1 - F_{\theta^*}^c)^2 - \sum_{\substack{i=1\\i\neq c}}^K\sum_{\substack{j=1\\j\neq i,c}}^K F_{\theta^*}^i F_{\theta^*}^j.$$

Since $0 \leq F_{\theta*}^j \leq (1 - F_{\theta*}^c)$ for $1 \leq j \leq K, j \neq c$ and $\sum_{\substack{j=1 \\ j \neq c}}^{K} F_{\theta*}^j = 1 - F_{\theta*}^c$, the above equation is bounded as

$$\left(\frac{K}{K-1}\right)(1 - F_{\theta*}^c)^2 \leq \sum_{k=1}^{K} \left(F_{\theta*}^k - y^k\right)^2 \leq 2\left(1 - F_{\theta*}^c\right)^2. \tag{14}$$

The minimum occurs when $F_{\theta*}^j = (1 - F_{\theta*}^c)/(K-1)$ for $1 \leq j \leq K, j \neq c$ and the maximum occurs when all the remaining probability (i.e., $1 - F_{\theta*}^c$) is given to one class besides the true class $c$, and the rest of the classes are assigned with zero probability. By applying the first order Taylor expansion for logarithm we have

$$-\log(F_{\theta*}^c) \approx 1 - F_{\theta*}^c,$$

where following the inequality in Equation (14) we have

$$\sqrt{\frac{1}{2}\sum_{k=1}^{K}\left(F_{\theta*}^k - y^k\right)^2} \leq 1 - F_{\theta*}^c \leq \sqrt{\frac{K-1}{K}\sum_{k=1}^{K}\left(F_{\theta*}^k - y^k\right)^2}.$$

Intuitively, the upper bound above is preferable in practice, because we would like to be less confidant in assigning probabilities to wrong classes. If we take the expectation of the above inequality because of Jenson's inequality, the upper bound is upper bounded by $\sqrt{\frac{K-1}{K}}\sqrt{L_{\text{MSE}}}$. However, here we consider the worst case approximation and approximate $1 - F_{\theta*}^c$ with the lower bound above. In practice, we observe that this approximation holds, and the network overfits very confidently and assigns the probability $1 - F_{\theta*}^c$ to a wrong class and zero to the remaining classes. Hence, by roughly approximating the expectation of a squared root with the squared root of expectation, we have

$$L \approx \sqrt{\frac{L_{\text{MSE}}}{2}}.$$

## 8.4 COMPUTATION OF EQUATION (9)

Consider a feedforward FC with ReLU activation function where i.i.d. zero mean random noise $\varepsilon_x$ with variance $\sigma_{\varepsilon_x}^2$ is added to the input. Then, assuming the output noise entries are independent from each other, we have

$$S = \frac{1}{K^2}\sum_{k=1}^{K}\text{Var}\left[\varepsilon_y^k\right] = \frac{1}{K^2}\sum_{k=1}^{K}\mathbb{E}\left[(\varepsilon_y^k)^2\right].$$

If we have $M$ hidden layers with $H_l, 1 \leq l \leq M$ units per layer, assuming the parameters are i.i.d. and independent from the input noise $\varepsilon_x$, and are drawn from the standard normal distribution, following the same computations as in Equation (11) for a network with $M$ hidden layers, $D$ input units and $K$ output units,

$$S = \frac{1}{K^2}\sum_{k=1}^{K}D\sigma_{\varepsilon_x}^2\prod_{l=1}^{M}\frac{H_l}{2}.$$

If all the hidden layers have the same number of units, $H_1 = H_2 = \cdots = H_M = H$, then,

$$S = \frac{D}{K}\left(\frac{H}{2}\right)^M \sigma_{\varepsilon_x}^2.$$

## 8.5 CIFAR-10 EXPERIMENTS

Figure 5 presents the effect of different initialization techniques, and of adding dropout and batch normalization layers to fully connected and convolutional neural networks trained on 1000 samples of the CIFAR-10 training dataset, and evaluated on the entire CIFAR-10 testing dataset. We observe again the strong relation between sensitivity $S_{\text{after}}$ and generalization error $L$ and the effect of these techniques on both $S_{\text{after}}$ and $L$. In Figure 6, we present the empirical results on the relation between $var$ defined in Equation (12) and $S$ defined in Equation (2). We experiment for 5 cases, where we change the second moment of the input $\sigma_x^2$ and the input noise $\sigma_{\varepsilon_x}^2$. In Figures 6(a) and 6(b), the original CIFAR-10 images are considered and in Figures 6(c), 6(d) and 6(e), we normalize the inputs accordingly. In all figures, the empirical relation between $var$ and $S$ shows a good match with Equation (13). We further experiment on image classification tasks for the MNIST and CIFAR-100 datasets in Appendix 8.6. We also provide numerical experiments for a regression task for the Boston house price prediction in Appendix 8.7.

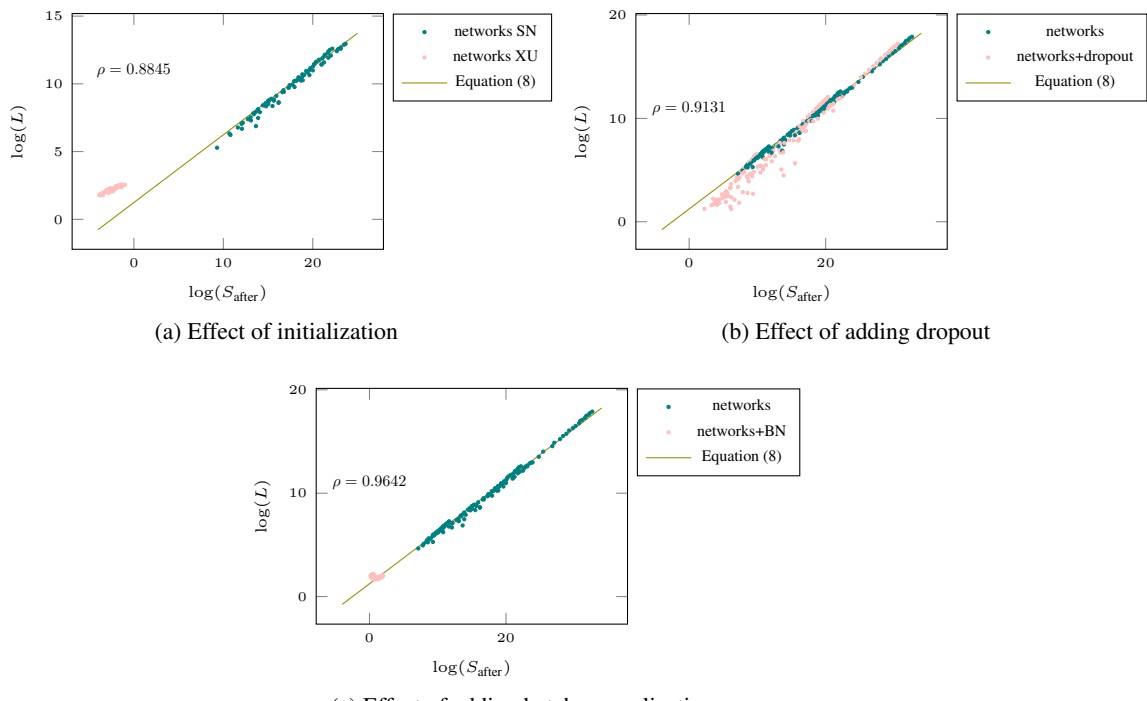

(a) Effect of initialization

(b) Effect of adding dropout

(c) Effect of adding batch normalization

Figure 5: Test loss versus sensitivity for networks trained on 1000 samples of the CIFAR-10 training dataset presenting the effect of initialization, dropout and batch normalization. Each point is the average over multiple runs and indicates a different architecture. (a) The networks are 5 layer FC, 2-4 layer CNN where the parameters are initially drawn from either Xavier uniform distribution (XU) or standard normal distribution (SN). (b) The networks are 3, 5, 7 layer FC and 1-4 layer CNN. The top most right pink point is the same network architecture as the top most right teal point when dropout is added to the configuration. Hence, for all network architectures we observe a shift of the numerical points towards bottom left of the figure when dropout is applied. (c) The networks are 3, 5, 7 layer FC and 1-4 layer CNN. In (b) and (c) the networks parameters are initially drawn from the standard normal distribution.

## 8.6 MNIST AND CIFAR-100 EXPERIMENTS

In this section, we present the experimental results for networks trained on 6000 samples of the MNIST[6] training dataset and evaluated on the entire MNIST testing dataset. Figures 7(a) and 7(b) show the results for fully connected neural networks with different number of layers and units and using regularization techniques batch normalization and dropout. In Figures 7(c) and 7(d), the results for convolutional neural networks are presented. Finally, in Figures 7(e) 7(f), the results on the comparison of the sensitivity of untrained networks $S_{\text{before}}$ with the test loss $L$ after the networks are trained, are presented. Figure 8 presents the sensitivity $S$ versus the loss $L$ for networks trained on the 1000 samples of the CIFAR-100 dataset[7]. The results on these two datasets are consistent with the rest of the paper, and once again we observe the relation between sensitivity and generalization and the effect of state-of-the-art techniques on both sensitivity and generalization.

## 8.7 REGRESSION TASK AND MSE LOSS

In this section, we investigate the relation between sensitivity and generalization error for regression tasks with mean square error criteria. The loss function in this setting is defined as

$$L_{\text{MSE}} = \mathbb{E}_{\theta^*}[L_{\theta^* \text{MSE}}] = \mathbb{E}_{\theta^*}\left[\mathbb{E}_{(x,y) \sim \tau_v}\left[(f_{\theta^*}(x) - y)^2\right]\right].$$

where $\theta^*$ is found by minimizing the mean square error on training dataset $\tau_t$ using the stochastic learning algorithm $\mathcal{A}$. Note that in this setting the output is the last layer of the neural network, the softmax layer is not added in this setting and the output layer has 1 unit, i.e., $K = 1$ and $y$ is a scalar. The bias and variance term are

---

[6] http://yann.lecun.com/exdb/mnist/
[7] https://www.cs.toronto.edu/~kriz/cifar.html

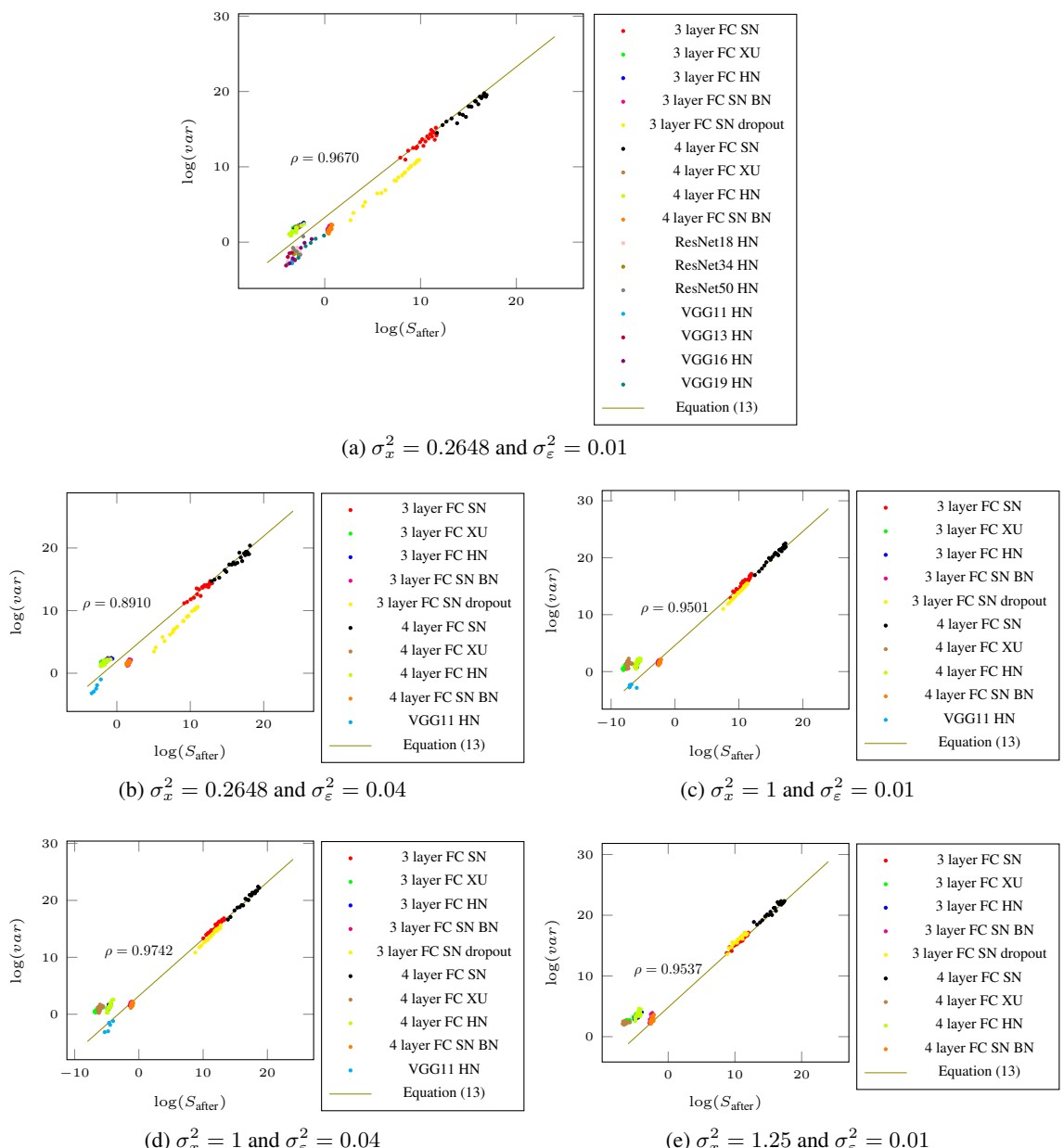

Figure 6: $var$ (Equation (12)) versus $S$ (Equation (2)) for networks trained on 1000 samples of the CIFAR-10 training dataset. The non-linearity is ReLU, and $\chi$ is neglected in the computation of Equation (13) in the figures. (a), (b) The non-normalized original CIFAR-10 input images. (c), (d) Normalized input images with zero-mean and unit variance. (e) We normalize the inputs to have unit variance and the mean is kept the same as the original images.

defined as

$$\varepsilon_{\text{bias}} = \mathbb{E}_{x,y}\left[\left(\mathbb{E}_{\theta^*}\left[f_{\theta^*}(x)\right] - y\right)^2\right],$$

and

$$\varepsilon_{\text{variance}} = \mathbb{E}_x\left[\text{Var}_{\theta^*}\left(f_{\theta^*}(x)\right)\right],$$

respectively. We consider the Boston housing dataset[8] where the objective is to predict the price of a house given 14 features (including crime rate, distance to employment centers, etc.). Figure 9 shows the results for comparing sensitivity and test loss among fully connected neural networks with 3-8 layers and 100-500 hidden units per

---

[8] https://www.cs.toronto.edu/~delve/data/boston/bostonDetail.html

layer; the networks are trained on 70% of the dataset and then evaluated on the remaining 30%. The networks could not get to zero training loss, so we stopped the training after 1000 epochs. The results are consistent on the relation between sensitivity $S$ and generalization error which for the regression task is $L_{\mathrm{MSE}}$ (Figure 9(e)). For a more detailed view, we observe that sensitivity is related to the variance in the bias-variance decomposition of the MSE loss (Figure 9(d)), and the MSE loss is the summation of the bias term and the variance term (Figure 9(c)).

## 8.8 DISCUSSION REGARDING $\varepsilon_{\mathrm{BIAS}}$

In this section, we present the results comparing $S_{\mathrm{after}}$ vs. $L$ when the networks are trained on different numbers of training samples (Figures 10(b) and 10(c)) and at different stages of training (Figure 10(d)) to validate the approximation made in Equation (8) where we neglect $\varepsilon_{\mathrm{bias}}$ and $\varepsilon_{\mathrm{noise}}$. Figure 10(a) shows sensitivity versus loss after the networks are trained on a subset of 1000 samples of the CIFAR-10 training dataset. Clearly, the approximations made in the computation of Equation (8) do not hold for ResNet18 and ResNet34 networks and the match is poor, contrary to the other part. Figure 10(b) shows that this problem can be explained (at least in part) by training the networks with more samples. For instance, in Figure 10(b) the yellow mark is ResNet18 with s=1, and the green mark is Resnet34 with s=0.5, and the transition of the results, towards the linear relation between $\log S$ and $\log L$, is clearly observed as we add more training data samples in the training process. Therefore, the larger the number of training samples, the better the approximation $\varepsilon_{\mathrm{bias}} \approx trainloss$ becomes and $\varepsilon_{\mathrm{variance}}$ becomes the dominant term in the test loss. Figure 10(d) shows the effect of the stage of the training. Both the sensitivity $S$ and the loss $L$ are computed at different stages of training (with some abuse of notation for $L$). In this figure, we observe the effect of the stage of training on the approximations made for computing Equation (8). At initial stages of the training, the bias $\varepsilon_{\mathrm{bias}}$ and noise $\varepsilon_{\mathrm{noise}}$ term in the test loss cannot be neglected.

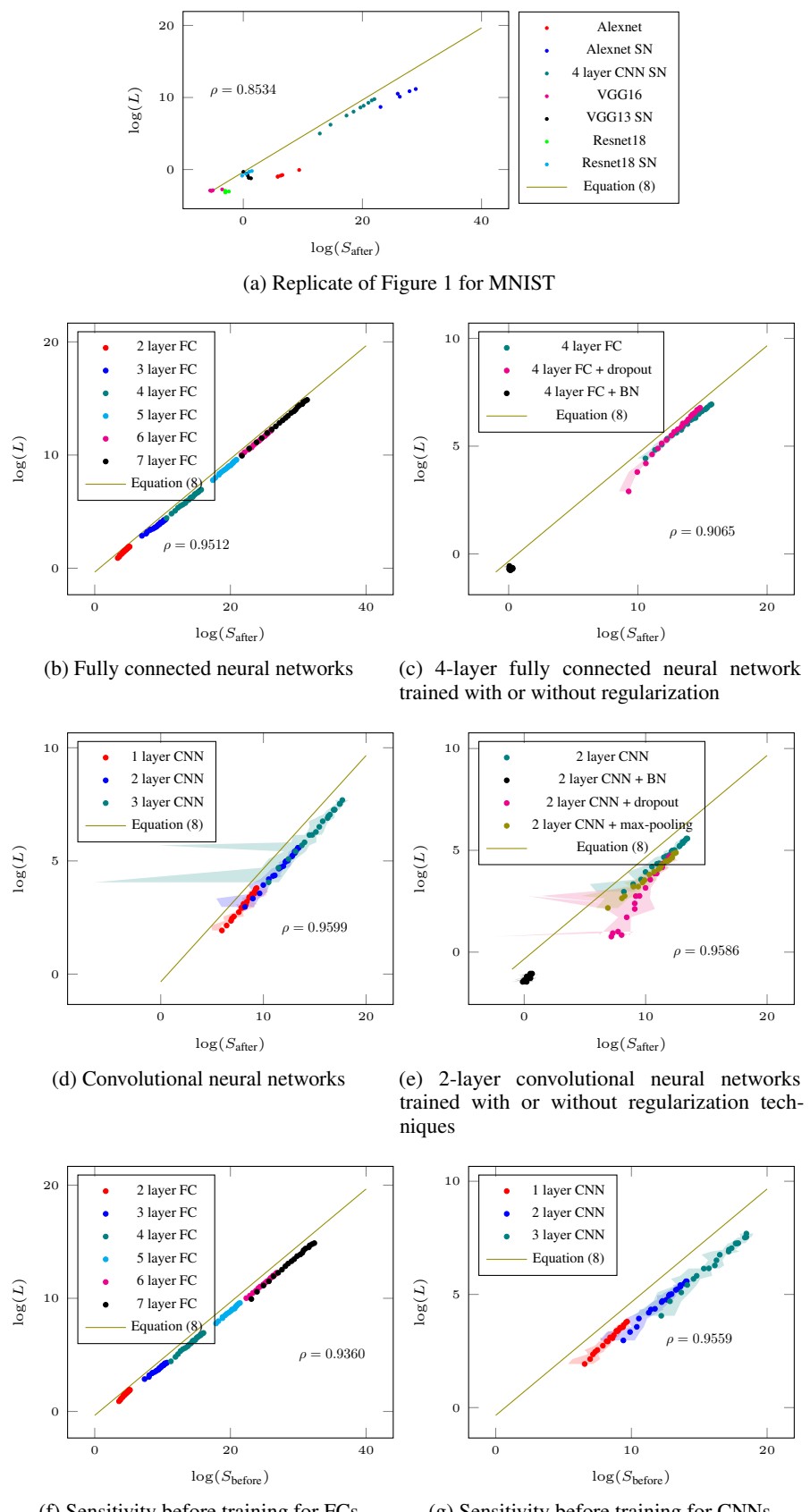

(a) Replicate of Figure 1 for MNIST

(b) Fully connected neural networks

(c) 4-layer fully connected neural network trained with or without regularization

(d) Convolutional neural networks

(e) 2-layer convolutional neural networks trained with or without regularization techniques

(f) Sensitivity before training for FCs

(g) Sensitivity before training for CNNs

Figure 7: Test loss versus sensitivity for networks trained on 6000 samples of the MNIST training dataset. Each point indicates a network with a different width and the sensitivity and test loss are averaged over multiple runs.

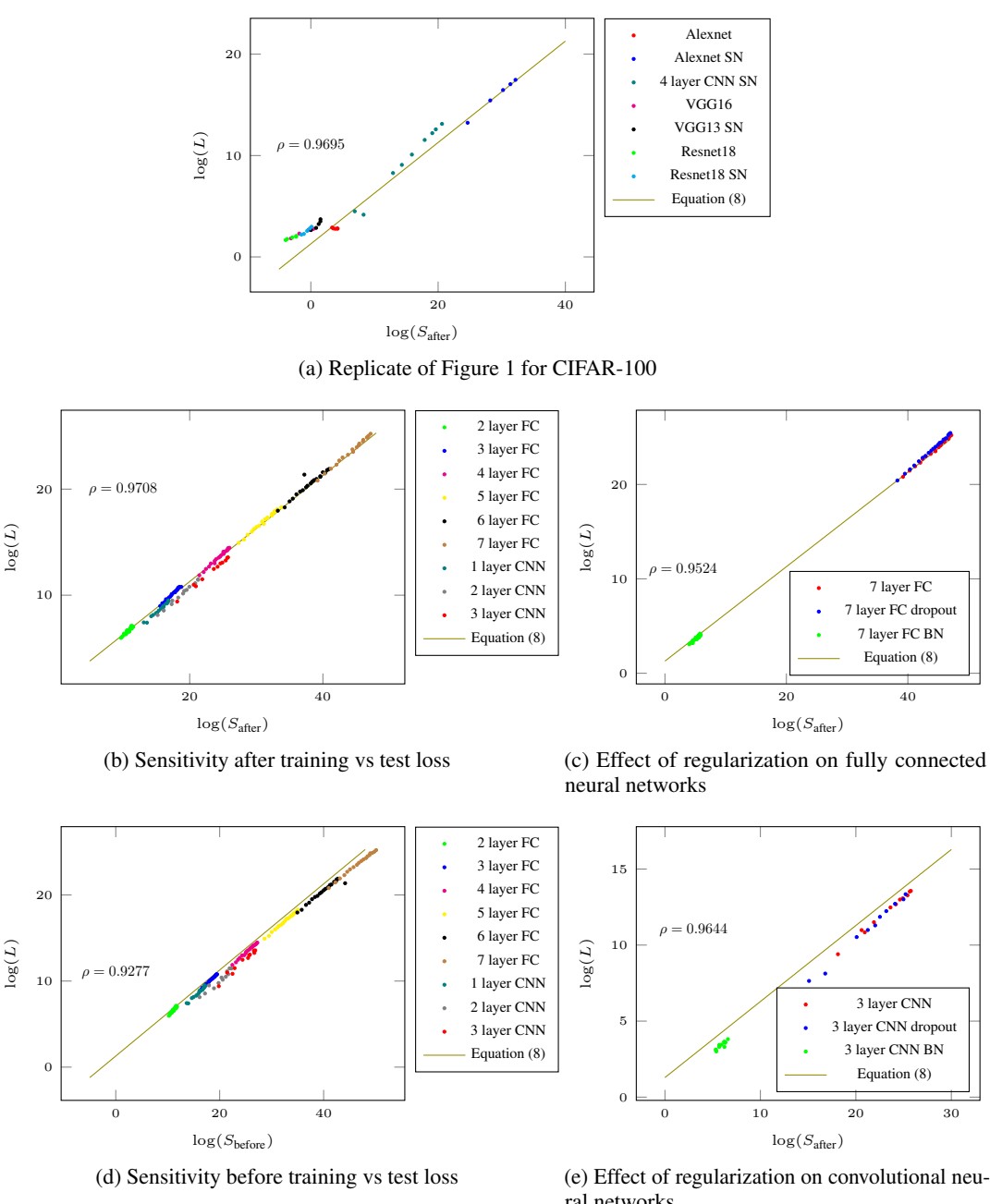

(a) Replicate of Figure 1 for CIFAR-100

(b) Sensitivity after training vs test loss

(c) Effect of regularization on fully connected neural networks

(d) Sensitivity before training vs test loss

(e) Effect of regularization on convolutional neural networks

Figure 8: $L$ versus $S$ for networks trained on 1000 samples of the CIFAR-100 training dataset. Each point indicates a network with a different width and the sensitivity and test loss are averaged over multiple runs.

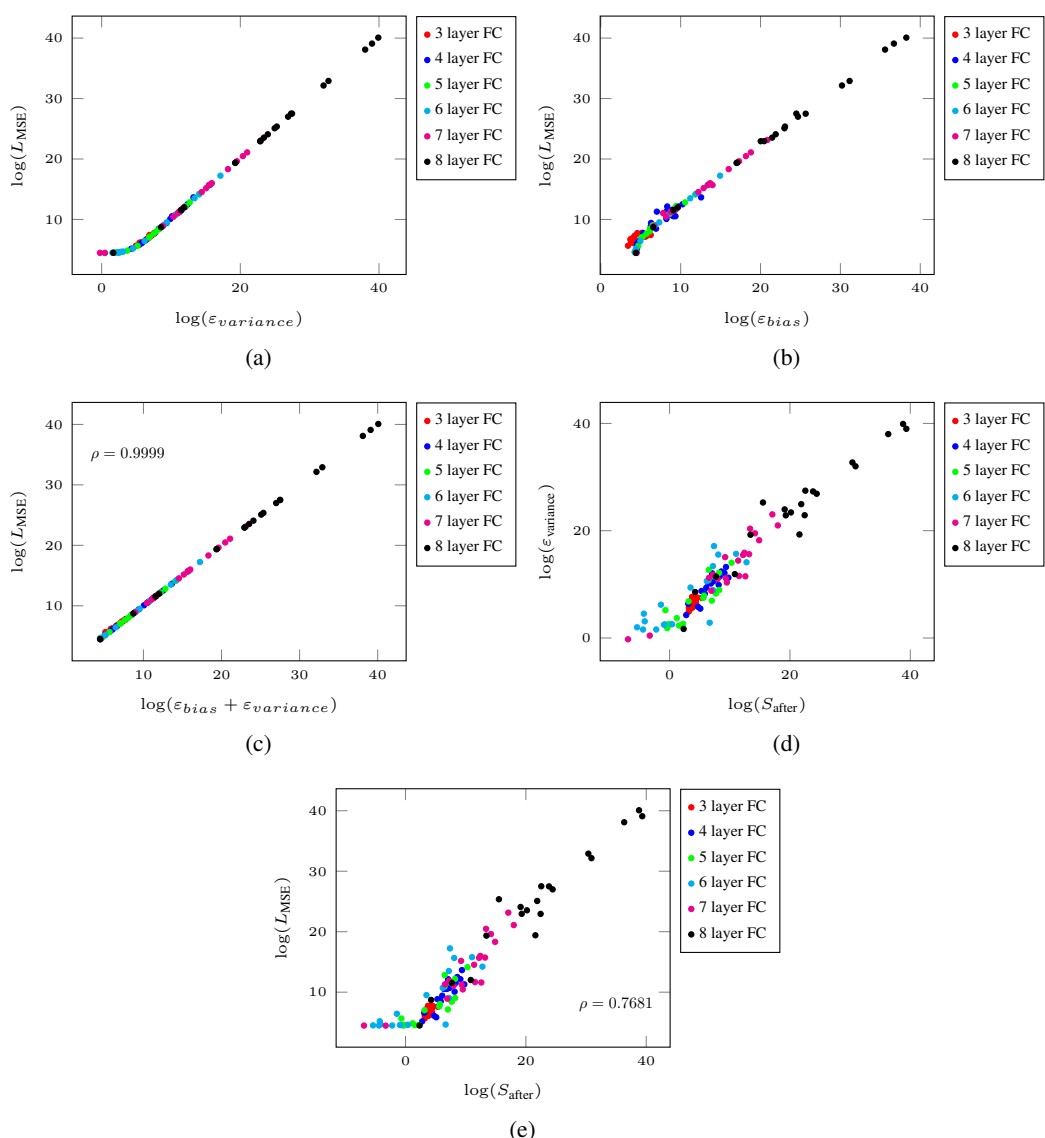

Figure 9: Variance, bias and sensitivity versus test loss for a regression task using the MSE loss. The fully connected neural networks are trained and evaluated on the Boston house price dataset. Each point indicates an average over multiple runs of a network with different widths $H$.

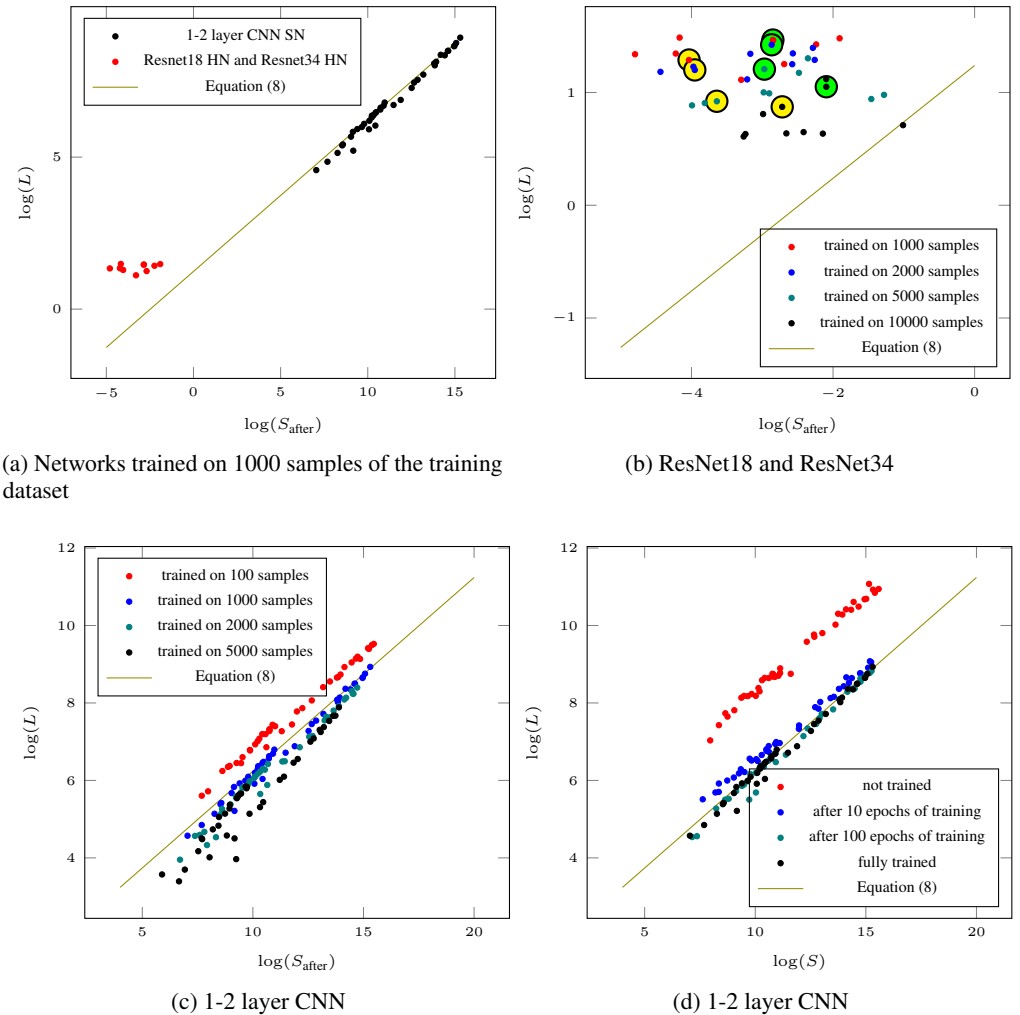

(a) Networks trained on 1000 samples of the training dataset

(b) ResNet18 and ResNet34

(c) 1-2 layer CNN

(d) 1-2 layer CNN

Figure 10: Sensitivity versus test loss for networks at different stages of training and trained on different numbers of training samples. Each point indicates an average over multiple runs of a network with a different width and depth. (b) is the zoom in of (a) on the bottom left, and we add the results of the same networks trained with different numbers of samples. In (b) The network parameters are drawn from a normal distribution by using the He technique. (c) and (d) are the zoom in of (a) on the top left, and we add the results for the same networks trained with different number of training samples in (c) and at different stages of training in (d). In (c) and (d) the network parameters are drawn from the standard normal distribution.

