# OpenReview forum: "On the Reflection of Sensitivity in the Generalization Error"
_ICLR.cc/2020/Conference — Reject_

### Official Review · AnonReviewer2 · 2019-10-23
**Official Blind Review #2**

**Rating:** 3

**Review:**

This paper studies the connection between sensitivity and generalization where sensitivity is roughly defined as the variance of the output of the network when gaussian noise is added to the input data (generated from the same distribution as the training error).

The paper is well-written and the experiments are very comprehensive. There are however 3 major issues with the current approach:

1- Novelty: Novak et al. 2018 suggests a very similar notation of sensitivity and they show correlation with generalization. Even though the authors site this work, they don't discuss the connection very clearly. In light of that work, there is very limited novelty in this paper.

2- Definition of test loss: Authors define the test loss to be cross-entropy but in almost all these tasks, what we care about is the task-loss which is 0/1 classification error on the test data and not the cross-entropy loss. These two loss behave very differently. In particular, the cross-entropy loss is very sensitive to the of variance of the output while 0/1 classification loss does not depend on it. Therefore, it is not surprising that there is high correlation between the output variance and the cross-entropy loss but it is not clear if this has anything to do with the test error.

3- Using test data in the complexity measure: The goal of understanding generalization is not just to get correlation with the test error. One can always use a validation set to get a very good correlation. Even when we have limited data, we can always put a small portion of the data for validation without loosing much in the final performance. The main goal is to predict generalization without using any access to the distribution. In particular, we need properties that show how networks behave on new data instead of simply measuring a property on the new data. Therefore, using a measure that is evaluated on new data is not really helpful.


********************************

After author rebuttals:

Authors have addressed one of my concerns (no 3) but the other two concerns are not addressed adequately. I increase my score to "weak reject" but not higher because of my concern about the novelty of the work in light of Novak et al. 2018.

**Experience Assessment:**

I have published in this field for several years.

**Review Assessment: Checking Correctness Of Derivations And Theory:**

I assessed the sensibility of the derivations and theory.

**Review Assessment: Checking Correctness Of Experiments:**

I carefully checked the experiments.

**Review Assessment: Thoroughness In Paper Reading:**

I read the paper thoroughly.

---

> ### Author Response · Authors · 2019-11-13
> **Reply to Reviewer #2**
>
> Thank you very much for your structured feedback which we will use in our response.
>
> 1- Novelty: Please refer to the reply to Reviewer 3, where we very thoroughly compared our work with [1] and pointed out, in particular, the differences between the two works.
>
> 2- Definition of the test loss: We would like to comment on this point from three aspects:
>
> - We find it quite surprising to find a relationship between the cross-entropy loss (which depends on labels of the input data) and the sensitivity of the neural network output (which does not depend on the labels of the input data). In particular, please refer to Equation (8) where the left-hand side depends on the labels and the right-hand side does not depend on the labels.
>
> - We can combine the bound found in [2] (refer to proposition 1 in [2] where the classification error is upper bounded by four times the regression loss) and the relation between cross-entropy loss and mean square error found in our work (Section 8.3), and find an upper bound for the classification error as a function of the cross-entropy loss.
>
> - The relation between the variance term in the bias-variance decomposition and the sensitivity can be extended to any loss with such a decomposition. [3] defines the concepts of bias, variance, and noise for the multi-class classification error, but there is still no rigorous multi-class classification error decomposition (which might not be purely additive). If such a decomposition is found for the classification error, then the relation between sensitivity and classification error would follow.
>
> 3- Using test data in the complexity measure: Thanks for pointing this out. This is exactly why the sensitivity metric is intriguing since this is exactly what S is in principle doing. The sensitivity metric is a property of the network and not of the input data. Although in the experiments presented in the paper S is computed on the testing data points, its value on train data exactly matches that on test data, and the same conclusions can be made for both. So, as long as the unseen data follows the same distribution as the accessed training data, the link with the generalization loss remains the same.
>
>
> [1] Novak, Roman, et al., "Sensitivity and generalization in neural networks: an empirical study." arXiv preprint arXiv:1802.08760 (2018).
> [2] Brady Neal, Sarthak Mittal, Aristide Baratin, Vinayak Tantia, Matthew Scicluna, Simon Lacoste-Julien, and Ioannis Mitliagkas. A modern take on the bias-variance tradeoff in neural networks. arXiv preprint arXiv:1810.08591, 2018.
> [3] P. Domingos and G. Hulten. A unified bias-variance decomposition and its applications. In Proceedings of the 17th International Conference on Machine Learning, pages 231–238, 2000.

---

> > ### Comment · AnonReviewer2 · 2019-11-13
> > **Thank you for your reponse**
> >
> > Thanks for your response. I still believe that the connections to Novak et. al. should be discussed clearly in the paper. I also believe that the novelty of the paper is very limited since there is little difference between what is suggested in Novak et. al. and this paper. Having experiments on ConvNets is great but is not enough for a paper to be accepted at this conference. About the test loss, I agree with authors' response but that is only analytical and what I really care is empirical correlation with 0/1 test error instead of the cross-entropy loss. Finally you have mentioned that "Although in the experiments presented in the paper S is computed on the testing data points, its value on train data exactly matches that on test data". This is not clear to me at all and that is why I encourage you to calculate and report this measure on the training set.

---

> > > ### Author Response · Authors · 2019-11-15
> > > **Sensitivity on the training set**
> > >
> > > Thanks for your suggestion, we will add in the paper the differences with [1], and give a short summary of the mentioned points.
> > >
> > > Please find below the calculated sensitivity on the MNIST train data set for a few trained networks (4 layer FC, 4 layer CNN, and VGG13 with various widths and number of channels). Since sensitivity is not using labeled data and the distribution of the input images are the same in test and train, the sensitivity value is the same when calculated on either set.
> > >
> > > S_test = [17627.77, 69103.00, 176774.00, 362343.86, 670139.63, 1087607.93, 148859.97, 2159199.86, 14964266.50, 42166734.79, 105627866.40, 221379445.12, 645672613.35, 1055347918.57, 1957933949.95, 2.54, 5.41, 18.52, 24.32]
> > >
> > > S_train = [17845.30, 69081.24, 176801.36, 362165.64, 663971.02, 1090227.46, 145624.26, 2184448.63, 15247152.58, 42963099.05, 105478721.26, 221472995.39, 652898741.97, 1052446483.39, 1965431193.4045093, 2.57, 5.65, 19.58, 26.77]
> > >
> > > The Pearson correlation coefficient between the two = 0.9999925
> > >
> > >
> > > We also plotted these on the same figure versus the generalization loss:
> > > https://ibb.co/JQKW9Cm

---

### Official Review · AnonReviewer3 · 2019-10-24
**Official Blind Review #3**

**Rating:** 3

**Review:**

This paper examines generalization performance of various neural network architectures in terms of a sensitivity metric that approximates how the error responds to perturbations of the input. A crude argument is presented for how the proposed sensitivity metric captures the variance term in the standard bias-variance decomposition of the loss. A number of experimental results are presented that show strong correlation between the sensitivity metric and the empirical test loss.

Understanding the distinguishing characteristics of networks that generalize well versus networks that generalize poorly is a central challenge in modern deep learning research, so the topic and analyses presented in this paper are salient and will be of interest to most of the community. The experimental results are intriguing and the presentation is clear and easy to read. While some may object to the egregious simplifications utilized in "deriving" the sensitivity metric, I believe this kind of analysis should be welcomed if it produces new insights and helps explain otherwise opaque empirical phenomena. All told, if taken in isolation from prior work, I think the insights and empirical results presented in this paper are quite interesting and certainly sufficient for acceptance to ICLR.

However, there is significant overlap with prior work that severely detracts from the novelty of the results presented here, and I think the community is already familiar with the paper's main conclusions. From the empirical viewpoint, [1] performs a very similar (and actually quite a bit more thorough) analysis, and reaches very similar conclusions. The authors do cite [1], but unless I missed something, their main argument for uniqueness is basically "in experiments, we prefer S to the Jacobian, because in order to compute S it is enough to look at the network as a black box that given an input, generates an output, without requiring further knowledge of the model." While this may be useful from the practical standpoint for some non-differentiable models, I'm not convinced that this distinction is really significant in terms of building insights or new understanding.

One additional way this paper is distinct from [1] is that it includes a theoretical "derivation" for the sensitivity metric. While I found the argument interesting, from the theoretical perspective, [2] gives much more rigorous and insightful arguments that help explain the observed phenomena.

Overall, I'm just not convinced this paper is novel enough to merit publication. But perhaps I've overlooked something, in which case I hope the author's response can highlight their unique contributions relative to prior work.

[1] Novak, Roman, et al. "Sensitivity and generalization in neural networks: an empirical study." arXiv preprint arXiv:1802.08760 (2018).
[2] Arora, Sanjeev, et al. "Stronger Generalization Bounds for Deep Nets via a Compression Approach." International Conference on Machine Learning. 2018.

**Experience Assessment:**

I have published one or two papers in this area.

**Review Assessment: Checking Correctness Of Derivations And Theory:**

I assessed the sensibility of the derivations and theory.

**Review Assessment: Checking Correctness Of Experiments:**

I assessed the sensibility of the experiments.

**Review Assessment: Thoroughness In Paper Reading:**

I read the paper at least twice and used my best judgement in assessing the paper.

---

> ### Author Response · Authors · 2019-11-13
> **Reply to Reviewer #3 (part 2/2)**
>
> Our work complements and reinforces the results presented in [1] by giving a new explanation in terms of sensitivity to the benefits of methods such as BN and dropout, and a new approximate tool to compare or pre-select different network architectures without requiring labeled data.
>
> Arora et al. [2] provides a very interesting generalization error bound based on metrics that allow noise stability (layer cushion, etc.), whereas our paper presents a direct link between noise stability (in terms of an average case local sensitivity measure) and the generalization loss.
>
> [1] Novak, Roman, et al., "Sensitivity and generalization in neural networks: an empirical study." arXiv preprint arXiv:1802.08760 (2018).
> [2] Arora, Sanjeev, et al. "Stronger Generalization Bounds for Deep Nets via a Compression Approach." International Conference on Machine Learning. 2018.
> [3] Yannis Dimopoulos, Paul Bourret, and Sovan Lek. Use of some sensitivity criteria for choosing networks with good generalization ability. Neural Processing Letters, 2(6):1–4, 1995.

---

> ### Author Response · Authors · 2019-11-13
> **Reply to Reviewer #3 (part 1/2)**
>
> We would like to thank the reviewer for their thorough reading of our paper.
>
> The intuition behind the correlation between sensitivity and generalization in neural networks goes back actually to 1995 [3]; even if that work was then limited to synthetic data. It wasn't until last year that Novak et al. [1] hinted on this correlation with an empirical study on fully connected neural networks in image classification datasets. They compare the sensitivity (as measured by the input-output Jacobian of the output of the softmax function) and the generalization gap (the difference between test and train accuracy) for trained feedforward fully connected neural networks with various depths, widths, and hyper-parameters, and conclude that the Jacobian norm is predictive of generalization depending on how close to the manifold of the training data the function is evaluated.
>
> The sensitivity S and the Jacobian J are indeed conceptually similar, but there are some minor differences between the metrics in the two papers. S is computed before the softmax layer whereas J is computed after the softmax layer. Because of the chain rule, J depends on the derivative of the softmax function with respect to the logits, which is very low for highly confident predictors (the ones which assign a very high probability to one class and almost zero to the other classes). For instance, if the predictor erroneously assigns a high probability to a wrong class, the derivative of the softmax function is very low, resulting in a very low J, so J might be misleading in this case as it would indicate good generalization. In contrast, the sensitivity S does not depend on the confidence level of the predictor.
>
> On the other hand, a practical motivation for using S instead of J is that in real-world applications where we are given multiple trained networks, the sensitivity metric S allows us to have an indication on the network architecture(s) with the best generalization ability without any access to their architecture, whereas computing J requires a backward pass and access to the architecture of the network.
>
> While [1] investigates the link between the norm of the Jacobian and the test error for feedforward fully connected neural networks, going beyond fully connected networks is needed in deep learning applications. Our work presents empirical results not only for convolutional networks but also for the state of the art neural network architectures such as VGGs and ResNets (Figure 1).
>
> Showing the correlation between sensitivity and generalization is only the first part of our work (Figure 1), which motivates the rest of the paper. The second and main part of the paper is to show how this relation sheds new light on understanding why certain techniques work in practice. In particular, we find a repeated link between the benefit of a large and diverse set of popular methods improving learning in deep-nets and the way they decrease S:
>
> - Batch Normalization (BN):
> There are quite a few results bringing insights on the reasons why BN works, here we provide an alternative explanation for the effectiveness of BN in terms of sensitivity (Figure 2). We further give a new viewpoint on the success of dropout and max-pooling. Networks with these methods have a lower sensitivity alongside with a lower generalization loss.
>
> - Initialization Techniques:
> Our work presents the impact of different initialization techniques and rediscovers the effectiveness of He and Xavier techniques by computing the sensitivity (Section 5.2, Figure 3).
>
> - Comparing Different Architectures:
> Our work presents empirical results on comparing depth versus width and convolutional versus fully connected networks (Section 5.1). From Figure 2 it is clear that the generalization ability of quite a few different architectures can be easily compared without the use of any labeled data, just by computing the sensitivity for the given trained architectures.
>
> - Sensitivity S of Untrained Networks:
> Our work compares the sensitivity of untrained networks and the generalization of trained networks in Section 6.1 (Figure 4). This result is important in architecture search since it hints at the generalization ability of the network before the networks are trained.
>
> Interestingly, despite the crude approximations in our derivations, there is a strong alignment between Equation (8) and the empirical results (Figure 1). Even when the match is loose, this relation suggests a convincing explanation (refer to Figure 10 in Section 8.8). Also, it is easily transferable to other machine learning tasks such as regression tasks (refer to Figure 9 in Section 8.7).

---

### Decision · Program_Chairs · 2019-12-19

**Decision:**

Reject

**Comment:**

The paper proposes a definition of the sensitivity of the output to random perturbations of the input and its link to generalization.

While both reviewers appreciated the timeliness of this research, they were taken aback by the striking similarity with the work of Novak et al. I encourage the authors to resubmit to a later conference with a lengthier analysis of the differences between the two frameworks, as they started to do in their rebuttal.